# Antimicrobial resistance, equity and justice in low- and middle-income countries: an intersectional critical interpretive synthesis

Katy Davis [1] ✉, Ralalicia Limato [1], Meenakshi Monga [1], Beatrice Egid[1], Sneha Paul [2], Susan Okioma[3], Owen Nyamwanza[4], Abriti Arjyal[5], Syeda Tahmina Ahmed [2], Ayuska Parajuli[5], Mavis Pearl Kwabla [6], Bachera Aktar [2], Anne S. W. Ngunjiri [3], Kate Hawkins[7], Russell Dacombe [1], Syed Masud Ahmed[2], Mustapha Immurana[6], Jane Thiomi[3], Fidelis EY Anumu [6], Webster Mavhu [4], Lilian Otiso[3], Sabina Faiz Rashid [2], Sushil Baral[5], Margaret Gyapong[6], Sally Theobald[1] & Rosie Steege[1]

Global inequities in the burden of antimicrobial resistance (AMR), between and within populations, are heavily influenced by the social and structural determinants of health. Yet, AMR action has had limited attention to equity, and social approaches to AMR haven't routinely gone beyond an exploration of knowledge and awareness around ABU. This represents a missed opportunity to design equitable interventions and policy across One Health. We report the results of a critical interpretive synthesis of the social and structural drivers of AMR in Low- and Middle-Income Countries and present a conceptual framework of these drivers, linking to the Sustainable Development Goals (SDGs). We emphasise the limitations of a biomedical dominance in AMR research, highlighting the value of wider bodies of evidence for understanding the drivers of AMR to support equity and justice. We argue AMR interventions need action across the SDGs to target the root causes and address significant gaps in evidence.

Antimicrobial resistance (AMR) occurs when bacteria, viruses, fungi and parasites evolve over time and are no longer susceptible to antimicrobials[1,2]. Common infections like urinary tract infections become life-threatening, increasing the risk of disease spread, illness and death. Some of the most significant resistant infections include carbapenem-resistant and cephalosporin-resistant bacteria, and rifampicin-resistant mycobacterium[3]. How antimicrobials are used has a significant impact on the risk of resistance developing. Low- and Middle-Income Countries (LMICs) experience up to 90 percent of total global deaths from AMR[4], high rates of infectious diseases, challenges in access to healthcare and global inequities relating to supply of antimicrobial access[5,6]. Research estimates that 250,000 deaths were

attributable to bacterial AMR in Africa in 2019[7]. South Asia, Latin America, and the Caribbean are forecasted to have the highest AMR mortality rate by 2050[8]. AMR burden is also unequally distributed within countries[9,10], but we do not have a detailed picture due to an absence of disaggregated AMR data[11].

The use of antimicrobials and the transmission of resistant infections are heavily influenced by the social and structural determinants of health - the non-medical factors that shape health outcomes and drive health inequities among and between populations[12] - and by broader geo-politics that drive global inequities in drug production and access, including the power of wealthy countries, a lack of accountability of pharmaceutical industries over antibiotic markets

[1]Department of International Public Health, Liverpool School of Tropical Medicine, Liverpool, UK. [2]BRAC James P Grant School of Public Health, BRAC University, Dhaka, Bangladesh. [3]LVCT Health, Nairobi, Kenya. [4]Centre for Sexual Health & HIV/AIDS Research (CeSHHAR), Harare, Zimbabwe. [5]HERD International, Bhaisepati, Lalitpur, Nepal. [6]Institute of Health Research, University of Health and Allied Sciences, Ho, Ghana. [7]Pamoja communications, Brighton, UK. ✉e-mail: katy.davis@lstmed.ac.uk

and a lack of democratic decision-making around global health[13]. Increasing evidence highlights that many of these factors are also driving antifungal resistance[14].

Known transmission routes for drug-resistant infections (DRIs) include human-to-human transmission, human–animal interaction, environmental exposure and contaminated food[1]. AMR is, therefore, often considered a quintessential 'One Health' issue in research and policy. However, One Health has been critiqued for a lack of engagement with the social sciences and inattention to issues of power[15,16]. It is understood that AMR is driven by many of the same processes that drive infectious diseases of poverty more widely, namely overcrowded living conditions, poor nutrition, lack of access to water, sanitation, and essential medicines and autonomy accessing healthcare[17]. Sustainable Development Goal (SDG) 3 recognises the need for substantial investment and focus on addressing these inequities and environmental factors for achieving global health goals[18].

The drivers of AMR therefore constitute a 'creeping disaster' – a term increasingly used to describe a complex, deep rooted and inequitable process that lacks definable temporal and spatial boundaries[19–21]. This framing draws on significant bodies of work on 'slow violence'[22,23] and 'environmental justice' that highlight the unequal distribution of environmental risks and benefits associated with diverse hazards, from pollution to infectious disease, and the need for accountability[24,25]. AMR, and relevant domains such as Water, Sanitation, and Hygiene (WASH), are therefore increasingly understood as part of a wider environmental injustice. As Murray et al.[26] argue, a One Health approach that explicitly embeds environmental justice will be more impactful in addressing the confluence of human, animal and environmental health as it explicitly addresses the role of societal inequities in influencing each.

It is well understood that inequitable power relations have a profound impact on health systems, opportunities and disease burden[27,28]. Nevertheless, much AMR research and policy frames AMR as a problem of antibiotic "misuse" and proposes addressing this through educational interventions targeted at specific groups, or stronger regulation on the sale of antibiotics[29,30]. This represents a critical missed opportunity to better tailor AMR interventions to address the root causes of AMR spread and therefore improve programmatic effectiveness, equity and sustainability[31]. We aim to counteract this narrative by highlighting that it is just as necessary to address these structural root causes of AMR. This might include improving access to safe housing and water[29], extending health coverage to rural and informal communities as well as addressing economic inequities and inequitable gender norms.

An intersectional feminist understanding of health inequity shifts the focus from isolated determinants of health, such as gender, age and race, towards an analysis of power. Intersectionality recognises that power operates '*simultaneously at intrapersonal, interpersonal, institutional and society-wide levels*'[32] (p1677). Intersectionality enhances understanding of not only who is vulnerable but also the complex processes underlying these risks[33] and therefore can inform policies, programmes and services to address these drivers of ill-health[17]. For example, research in Bangladesh identified specific occupational hazards leading to increased risk of urinary tract infections among women garment workers[34]. Given the high risk of AMR associated with urinary tract infections, this represents a specific AMR vulnerability at the intersection of occupation and gender, identified through the use of an intersectional lens. However, the use of intersectional approaches remains fairly new in the AMR field[35,36].

Intersectionality-informed analysis involves attention to the interactions between social determinants and the broader systems of oppression shaping health outcomes[33,37,38]. It is therefore aligned with a One Health approach, which considers the interconnectedness of humans, the environment and animals, though intersectionality brings a much needed focus on power. For example, intersectional

approaches have identified synergistic effects of age, ethnicity, poverty, livelihood, and gender that have led some to experience less positive impacts from health interventions than others, or greater barriers to accessing health services during pandemics[27,39]. Evidence points to how tradition, patriarchy, culture, gender norms, laws and social and structural factors affect women in particular, with women having lower status and less control over decision-making[40].

Recent reviews[10,36,41] have demonstrated the significant link between gender and AMR. Yet, these equity considerations are not yet reflected in policy making. For example, a recent review of 145 National Action Plans for AMR found that 125 of these did not include mentions of sex or gender[42]. This is an oversight that risks exacerbating these inequities. This also points to the significant evidence gaps that remain in research exploring inequities in AMR linked to root causes and context-specific processes and manifestations[31,35,36] and the need to synthesise this evidence for policy making.

In this paper we describe the results of a critical interpretive synthesis of the structural drivers of inequity in the context of AMR, with an intersectional lens. Structural processes are those that relate to the ways in which power and resources are shared within society at multiple scales and are the focus of a number of theories in the social sciences of health including the social and structural determinants of health[40,43], syndemic theory[44], intersectionality[33] ecosocial theory[45,46], health justice[47] critical medical anthropology[48] and health and human rights. These theories leverage grounded empirical methodologies to understand nuanced and context-specific realities of how health is impacted by various factors[49].

We use a critical interpretive synthesis (CIS) approach to review the diverse and disparate evidence on intersectional inequities driving AMR, and experiences of AMR, to synthesise and deepen our understanding of the processes shaping the inequitable burden, and present a conceptual framework to evidence this. Our review builds on recent CIS[24,50] that situate AMR policy and framings within wider frameworks of equity and justice, and which find that One Health-AMR governance responses do not explicitly integrate health equity concerns or considerations of the root causes driving AMR spread and increasing antimicrobial use (AMU). We sought to look beyond literature describing the inequitable burden of AMR and to identify literature that draws links between this inequitable burden and the broader structures of power. This understanding is necessary for designing equitable and effective interventions to address AMR and inform equitable policy making[24]. Building on this review of the literature, we develop and present a conceptual framework that moves beyond common framings of AMR as a problem of antibiotic "misuse" to illustrate the structural drivers of AMR spread and the ways that these are rendered invisible by current approaches to AMR surveillance.

## Results
### Overview and study characteristics
A total of 181 articles were included in the review (Fig. 1). The number of published articles reporting research on structural inequities as they relate to AMR remained steady over 2019-2024, with on average 30 articles published a year. Articles most frequently reported on research carried out in India ($n = 17$), China ($n = 16$), Uganda ($n = 15$), Kenya ($n = 13$), Tanzania ($n = 13$) and Ethiopia ($n = 12$) (see Fig. 2). Articles predominantly focused on human health (97%, $n = 176$). 13% ($n = 24$) focused on animal health and 8% ($n = 14$) focused on environmental health. 15% ($n = 27$) focused on the intersection of human, animal and environmental health.

### Findings
We present thematic findings and draw on intersectional theory to unpack the structural drivers of AMR with an attention to structural inequities and power relations.

 

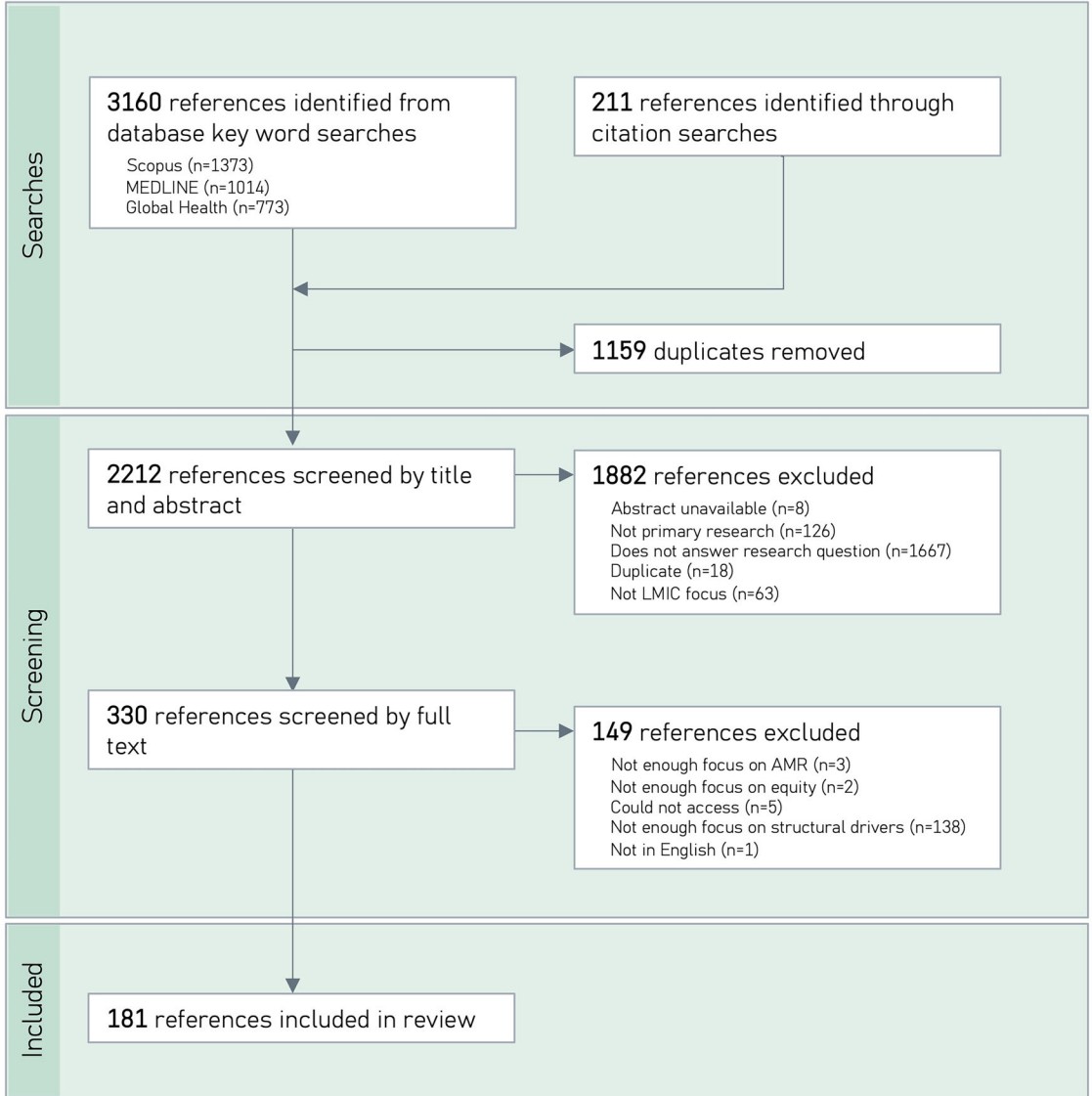

**Fig. 1 | PRISMA (Preferred Reporting Items for Systematic Reviews and Meta-Analyses) flowchart illustrating search and screening processes.**

## Susceptibility to infection

Biological susceptibility to infection is increased by malnutrition. This is particularly the case for drug-resistant tuberculosis (DR-TB), which a number of articles across Africa and Asia[51–53] highlighted as disproportionately affecting low income groups and women and girls, due to a combination of poverty and gender norms[54,55]. Malnutrition is known to be a symptom of inequitable food systems and distribution of resources at global, community and household level[56].

In some research contexts, DR-TB disproportionately impacts young women[57–59]. In Nigeria, higher rates of DR-TB among young women are reportedly due to the higher prevalence of HIV among this group[60], which increases susceptibility to DR-TB. Authors suggest that higher HIV rates are due to intergenerational relationships in which young women marry older men who are more likely to be living with HIV[60]. This highlights how gendered norms influence both exposure and susceptibility. Similarly, Girum et al.[61] highlight how inequitable gender power relations that drive sexual violence and limit women's reproductive rights lead to HIV among women.

## Access to vaccination and immunisation

Closely linked to susceptibility, inequitable vaccine access in India is highlighted as leading to inequitable susceptibility to resistant infections and infectious disease more widely[53]. Also in India, Kumar et al.[62] estimated that increasing childhood vaccination coverage can significantly reduce antibiotic demand among the poorest quintile of the population. Other studies propose vaccinations as of potential value among specific populations with high rates of resistant infections such as in urban informal settlements at risk of drug resistant salmonella in Kenya[63] and those in forest-going occupations at risk of resistant malaria in Cambodia[64].

## Exposure to infection and antimicrobials

Articles describe exposures to resistant infection and antibiotics primarily through the inequitable conditions of living and livelihoods associated with poverty and marginalisation, common among low-income communities and informal settlements. Overcrowding and associated lack of ventilation increase the spread of multidrug-resistant tuberculosis (MDR-TB)[51,53,59,65,66]. Limited or inequitable access to clean water and quality sanitation provides opportunities for proliferation of resistant bacteria[66–70]. For example, animal and human faecal contamination in soil in urban communities in Mozambique were linked to *E. coli* exposure from soil[71].

Livelihood opportunities are heavily influenced by socioeconomic and gendered inequities, which have inequitable ramifications on

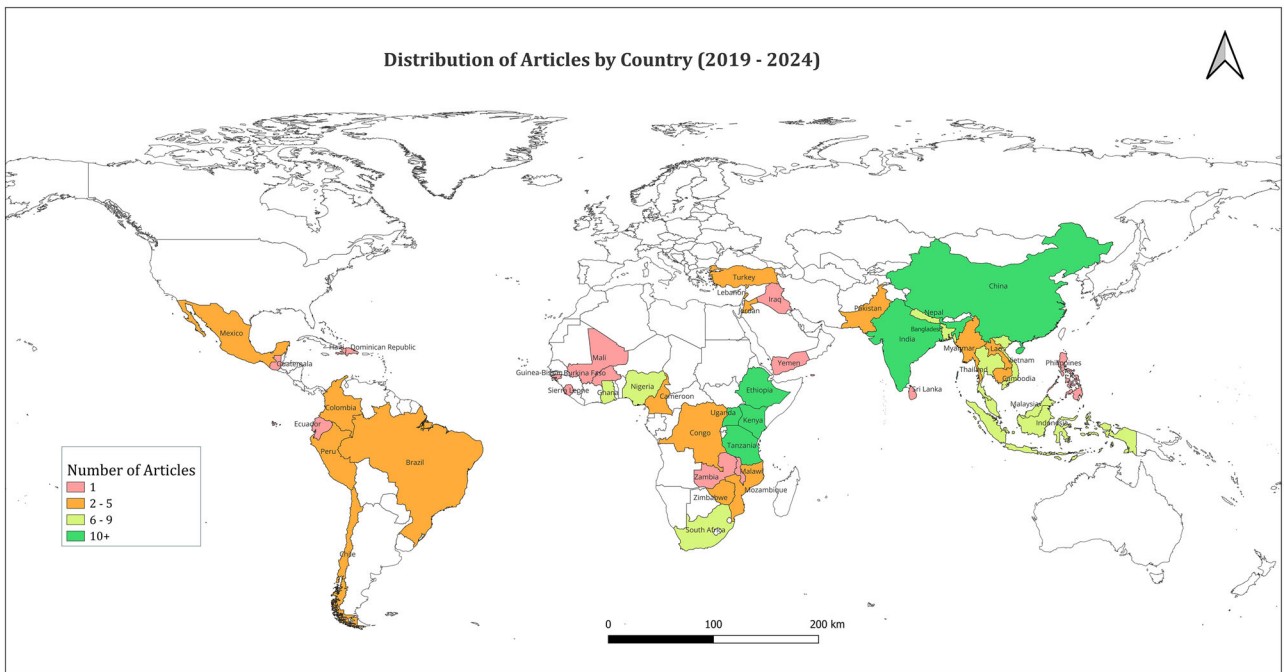

**Fig. 2 | Distribution of articles by country 2019–2024.**

exposure. Exposure to both infections and antibiotics used in animals is affected by livelihood, poverty and gender[72]. In rural communities in Cambodia, women are responsible for the care of poultry and pigs, among which carbapenemase- and extended-spectrum cephalosporinase-producing *Escherichia coli* and *Klebsiella pneumoniae* are significant, leading to gender-specific exposures to infection[73]. As women are also responsible for family caregiving, this can mean that children have high exposure. A study in Nepal found that gender norms dictate that women are involved in feeding and cleaning livestock but men are more likely to be in agricultural decision-making positions[74]. In Tanzania, Barasa[75] found that women and girls are responsible for administering medicines to sick animals and thus are more prone to exposure to infection and antibiotics. On the contrary, in Kenyan pastoralist communities, men are more exposed to antimicrobials and resistant infections because they take on the bulk of the work in direct contact with animals and use antibiotics extensively in the management of livestock disease[76]. One study reporting on melioidosis in Malaysia highlighted that men are more exposed to infection due to their involvement in soil-related occupations such as rice paddy farming and working on palm oil and rubber plantations[54]. Gendered forest-based livelihoods among some rural communities in Cambodia expose men, in particular, to repeated malaria infections and the subsequent treatments can lead to resistance[64].

Studies in Kenya, Brazil and Guinea-Bissau have identified that sex workers are at high risk of drug-resistant sexually transmitted infections such as gonorrhoea, syphilis and HIV, highlighting their occupational exposure to resistant infections[77–79]. Further, studies in Indonesia and Kenya reported that sex workers may be pushed to engage in activities that leave them more exposed, for example, having less bargaining power over condom use[80,81].

Cuboia et al.[58] describe high rates of DR-TB in areas of Mozambique close to the South African border. They suggest that this is due to the presence of high numbers of people who undertake seasonal mining work in South Africa, which increases their exposure to mining chemicals (such as silica dust) and poorly ventilated working conditions, combining to increase the risk of TB[82].

## Health seeking pathways

Articles described both the health seeking of those with resistant infections, as well as the aspects of health seeking and access that might facilitate the development of resistance to antimicrobials. These include barriers to accessing health services mediated by occupational and income-related issues, agency and gendered power relations, and geographical access.

Low income communities may not have finances to access formal health services or to ensure continuation of antimicrobial treatment for HIV or TB (which both require long-term courses), whether due to direct or indirect costs associated with seeking healthcare, and may need significant time to access cash required for seeking formal health care[83–85]. Hidden costs include transport and paying for out of stock medications[86,87]. Daily labourers may also have less time to seek care due to precarity of occupation, such as HIV-positive Mozambican migrants working in South Africa[88]. Poverty and precarious employment can lead to fluctuating ability to pay for medicine and can therefore be a barrier to continuation of treatment[89]. In Vietnam, unforeseen illness, crop loss or animal disease among ethnic minority communities create cyclical precarity and create further financial barriers to accessing health services[90].

Household power relations affect health seeking autonomy. Barasa and Virhia[83] report that in Tanzania, young boys' health is prioritised due to their herding duties, while girls and women must negotiate access to antimicrobials through older women in the house and then via men. Mothers in Northern Tanzania report lack of financial support from their husbands as a barrier to health seeking for their children[86]. In contrast, women garment workers in urban Bangladesh are reported to have significant health-seeking agency to access antimicrobials and other medicines due to their high rates of employment and ability to access drug shops[91].

Long distances and limited infrastructure pose barriers to accessing health services, and people with longer transport time to health services in Bangladesh have been seen to experience higher burden of AMR[92]. Access to health services is often linked to country-level structural inequities. Poor infrastructure, in terms of roads and diagnostic resources for identifying resistant infections were cited in rural

areas[54,83,93]. As Barasa and Virhia explain, poor infrastructure in pastoralist settings is *"a reflection of pastoralists' relative lack of power, living in geographic areas marginal to the national political process"*[83] (p 14).

Physical distance and lack of access to infection prevention methods can also impact mobile or migratory communities. Forest areas in Cambodia experience distinct malaria transmission characteristics due to specific forest vectors, climate and land use change, livelihoods, migration and poor health infrastructure[64]. Those with forest-going occupations at the Thailand/Cambodia border (a current epicentre of the emergence of drug- resistant *Plasmodium falciparum*) have limited access to mosquito nets and experience significant delays before they can access necessary antimalarial drugs[53], which can increase the risk of resistance.

Inadequate infrastructure and precarious working conditions also intersect to shape access to and use of antimicrobials in urban informal settlements but in different ways[85,94]. As Nabirye et al note, "everyday practices concerning antibiotic use in an informal settlement are entangled with precarious labour conditions, infrastructural deficiencies, and experiences of frequent illness"[95], highlighting how antibiotics are used to compensate for inadequate WASH and health system access[94].

Experiences and perceptions of treatment in health systems, including discrimination, affect likelihood of use. Barasa and Virhia[60] note that, for pastoralists in Tanzania, "religious bias, lack of medication, long waiting times and treatment costs render health facilities as places to be avoided during illness, rather than places where they can be healed" (p 14).

## Self-medication and informal prescription
Given the complex health-seeking pathways described above, and particularly the many barriers to accessing formal care, people engage in diverse non-formal routes to accessing antimicrobials. 'Self-medication' refers to the seeking of antimicrobials without prescription, either through informal providers, family or social networks, including in contexts where antibiotics are available to purchase without a prescription[96]. In articles included in our review, accessing antimicrobials through these routes is often described as a response to barriers to seeking formal healthcare described above. Among low-income populations, there is pressure to save time and money by accessing antimicrobials through non-formal prescription, which is seen to be more accessible and cheaper[96,97]. Parents and caregivers in Tanzania provide informally acquired antibiotics to children for these reasons[98]. Shukla et al[99] identified a high level of self-medication among rural dwellers due to cost and geographical access.

In pastoral communities in Tanzania, where men make decisions about how women access healthcare, men often purchased antibiotics without a prescription and brought them home to women with undiagnosed illnesses[54]. This could include stocking up with medicines during the drier months and using them to treat both people and animals during vector-borne disease outbreaks in the wet season. In Malawi and Zimbabwe, purchasing of antimicrobials from informal providers was often forced by drugs being out of stock in formal health systems[95] and in Ethiopia, anticipating a lack of antibiotics availability in public hospitals was a barrier to people investing expenses to attend[100]. This highlights the equity issues of health system resources.

Informal medication of livestock animals is often driven by the same processes. Pham-Duc et al[101] highlight the limited economic opportunities for livestock keepers in Vietnam and how prophylactic use of antimicrobials in animals was seen as important to guarantee their otherwise precarious livelihoods. Likewise, in northwestern China, antibiotic use for infection prevention in chicken farms was associated with lower income farms[102].

Additionally, mistrust in formal health systems or governments was often reported as a driver of self-medication. This was often due to previous experiences of formal sellers having stock-outs[90,103], the government's management of past disease outbreaks[104] general lack of confidence in formal health systems and health infrastructure[93], or previous experiences of neglect or discrimination[105].

## Formal prescription practices
Where patients can access formal diagnosis and prescription in hospitals and communities, prescription practices were reported to be shaped by economic pressures placed on both individual patients and prescribers including health system resourcing[68]. Prescribers in refugee camps in Lebanon face diagnostic uncertainty in low-resource settings, potentially leading to overprescription of pregnant women with urinary tract infections[106]. Pearson and Chandler[68] found that prescribing decisions were made in response to, and with awareness of, health-compromising environments in both Africa and Asia. Similar pressures were identified in South Africa and Sri Lanka[107]. In contexts where people cannot afford to purchase complete courses of antimicrobials, or where health services are under-resourced, prescribers describe dispensing incomplete courses of antimicrobials, such as low-income settings in Ghana[84].

In low-income contexts, financial pressures on prescribers and drug sellers also influence prescription practices. Caudell et al[108] report that agrovet shop owners across Southern Africa often have precarious livelihoods themselves, supported by research in Ethiopia[100].

## Treatment completion/continuity
Many of the same drivers of self-medication and barriers to accessing health services also influenced peoples' ability to complete courses of antimicrobials. Just as economic pressures and poverty are barriers to health seeking in the first instance, they can interrupt treatment[89]. Prescribers in Brazil reported that cuts in social protection interfered with continuation of treatment for MDR-TB[109]. Additionally in Brazil, Santos et al[69] highlight that food and transport expenses can limit continuity of treatment even when it's free. In South Africa and Uganda, food insecurity interrupted courses of treatment since MDR-TB treatment must be taken after a meal[110]. Migrant workers in Guangdong Province in China were more likely to be excluded from health insurance and their job precarity could affect their ability to pay for long-term MDR treatment[111].

Gender norms and relations also affect treatment continuity. In Pakistan, Abubakar et al[112] report that deep rooted gender discrimination limited women's health seeking agency and requires they attend hospitals with male relatives, often leading to discontinuation of TB treatment. Implicit gender biases within households affect whether girls complete courses of antibiotics in Mali[113]. Women in Tanzania report that burden of household work can get in the way of administering medicines to children at the recommended times[86]. Women working in diverse informal work in Bengaluru, India, feared losing work from time spent seeking health care. This was a key barrier to continuing treatment for tuberculosis, heightened during the COVID pandemic[114].

Stigma and lack of social support could also significantly affect follow-up. Badgeba et al[51] report that in Ethiopia, stigma associated with TB reduces social support which contributes to treatment interruptions. This was echoed by Wekunda et al[115]'s findings in Kenya, that also highlighted challenges of distance and transport.

## Use of medication in animals
Fewer studies focused on the socioeconomic dimensions of AMR in animal health, but some of the above themes around health-seeking and self-medication also apply to animal health. Qualified veterinarians are often not located close to rural livestock-keeping communities[108], and Campbell et al[116] described women who keep chickens in Kenya as having less access to capital than men and are dependent on their

husbands to purchase veterinary products and feed. Work in Nigeria suggested that small scale poultry farmers did not consult veterinarians due to financial pressures[117]. The use of medications in animals can also, in turn, influence resistance in animal populations and lead to environmental pollution with antibiotics.

## Knowledge of AMR and antimicrobials

Articles linked access to formal education, occupational training and health messaging with knowledge of AMR[74,118,119] and socioeconomic status[120,121]. In particular, level of knowledge of AMR appeared linked to access to health facilities. Pauzi et al.[122] found that older research participants in Malaysia had a greater understanding of antibiotic use and resistance because they visited health facilities more. Subedi et al.[74] found that research participants above 45 years old had lower knowledge due to lower literacy and social media access, and Ha et al.[123] report that rural and ethnic minority communities in Vietnam – particularly those working in the agriculture, fishery or forestry sector – had low levels of knowledge about AMR, likely because they have low levels of access to health services and information. This points to a key intersection between access to formal health systems and access to knowledge of AMR. Additionally, Do et al.[96] found that poorly labelled medicines limited the agency of communities in Africa and Asia to make decisions about antibiotic use.

In rural Nepal, gender norms dictate that women are expected to be responsible as primary caregivers for knowing how and when to give medicines to children[124], and similar norms were identified in Northern Tanzania[86]. This was linked to women's access to female community health volunteers who are expected to provide health advice. In contrast, men are expected to seek health care via doctors and in this way these different spaces of knowledge sharing are highly gendered.

Knowledge of AMR and recommended antimicrobial use, therefore, factor into people's chosen health seeking pathways, self-medication practices, and the use of antimicrobials in animals. Articles highlighted that AMR knowledge inequities often intersect with other inequities that relate to access to healthcare[100].

## Experiences of care and impacts of infection

Several studies reported catastrophic costs, associated with direct and indirect health expenditures, from resistant infections impacting those living in poverty or with low incomes most significantly[125]. Kaswa et al.[126] reported a significant decrease in employment among people going through TB treatment in Democratic Republic of Congo, and increased reported poverty levels, food insecurity and interruption to schooling. Catastrophic costs were greater for those with DR-TB. Pham et al.[127] found that those with lower education and those unemployed at the beginning of MDR-TB treatment were more likely to face catastrophic costs, and for many, income did not bounce back at the end of treatment. Likewise, in Zimbabwe, Timire et al. found that DR-TB narrowed job opportunities even after completion of treatment[128]. In Brazil, this aspect of MDR-TB impacted those under 40 years most due to their breadwinning roles[125]. Socioeconomic stressors also led to reduced psychological wellbeing[127,129]. Taylor et al.[110] describe "a vicious downward spiral in overall well-being, affecting most severely those who were already worse off to begin with" in South Africa and Uganda (p 8).

People with MDR-TB in South Africa and Uganda report that disclosing diagnoses sometimes lead to compromising social support, the loss of relationship and even being expelled from home[130]. This stigma can be gendered, with women living with tuberculosis in India reporting increased stigma during the COVID-19 pandemic[114]. A loss of reputation among people with DR-TB in Vietnam was reported by Redwood et al.[131]. The stigma of drug-resistant diseases such as HIV or TB can also lead to non-disclosure and loneliness, as reported by HIV-positive Mozambican migrants in South Africa[88].

## Summary of results

Results highlight that intersectional power structures create unique and context-specific inequities that relate to exposure and susceptibilities to infection. Exposure to infection is driven by environments that drive resistance and spread of infections, whether these are living conditions, occupational working conditions or healthcare settings, and include contexts in which people are in close proximity to animals or environmental pollution. This exposure interacts with biological susceptibility (in turn influenced by vaccination) to create risk of acquiring infections, including resistant infections.

Health seeking pathways are shown to be characterised by both inequitable access to health services and inequitable experiences of formal and informal prescription and treatment completion. Complex and context-specific barriers to seeking timely and affordable formal health care result in people engaging in diverse approaches to accessing antimicrobials through self-medication and informal prescription. People also face context-specific and inequitable challenges and barriers to completing courses of medication and treatment, particularly where treatment periods are extended, whether they have received prescriptions through formal or informal routes.

## Conceptual framework

We build on the results of our review, the wider literature on the structural drivers of AMR, and our embedded knowledge as an international research consortium, to present a conceptual framework that highlights the complex and power-laden social processes along the AMR pathway from exposure to impact. Our review identified intersectional inequities that are significant drivers of AMR. These include inequities in: access to vaccination and immunisation, susceptibility to infection, exposure to infection, available health seeking pathways, the resulting self-medication, formal and informal prescription processes (including where medication is inaccessible altogether), the barriers to treatment continuity and completion and diverse experiences of care and impacts of resistant infection. These are illustrated in Fig. 3 as steps on the complex pathway of AMR in people's experience from exposure to impact. Many of these themes are relevant to human, animal and plant health, and access to knowledge of AMR and antimicrobials intersects with each. Underlying these are context-specific intersectional inequities in e.g., malnutrition, access to WASH, household decision-making, economic autonomy, work and livelihoods. Each of these links in various and multiple ways to the sustainable development goals which are identified in Fig. 3[1].

This framework highlights the limitations of relying solely on surveillance data collected at facility-level, where focus of AMR disease surveillance and policy is often placed. We highlight that AMR surveillance data, whilst critical, is just the 'tip of the iceberg'. It does not reflect the complex, diverse and differential reasons why people may not access a facility or may not be able to continue treatment, the inequities that define peoples' differential experiences, nor the structural power processes that are the root causes of these inequities. We link to the SDGs to show how action on AMR will require action across all the SDGs. This includes attention to conditions of living, access to water, sanitation and hygiene (WASH) systems, food insecurity, livelihoods in which people are exposed to harmful conditions and experience precarity, inequitable access to affordable and person-centred health systems, gender power relations, household decision-making and economic autonomy and inequities in access to education and training. We argue that more attentions in AMR funding and research needs to be paid to these foundations.

## Discussion

Our CIS contributes to a growing discussion around the structural root causes of inequities relating to the spread of AMR in LMICs, highlighting diverse and complex trends and processes that vary across

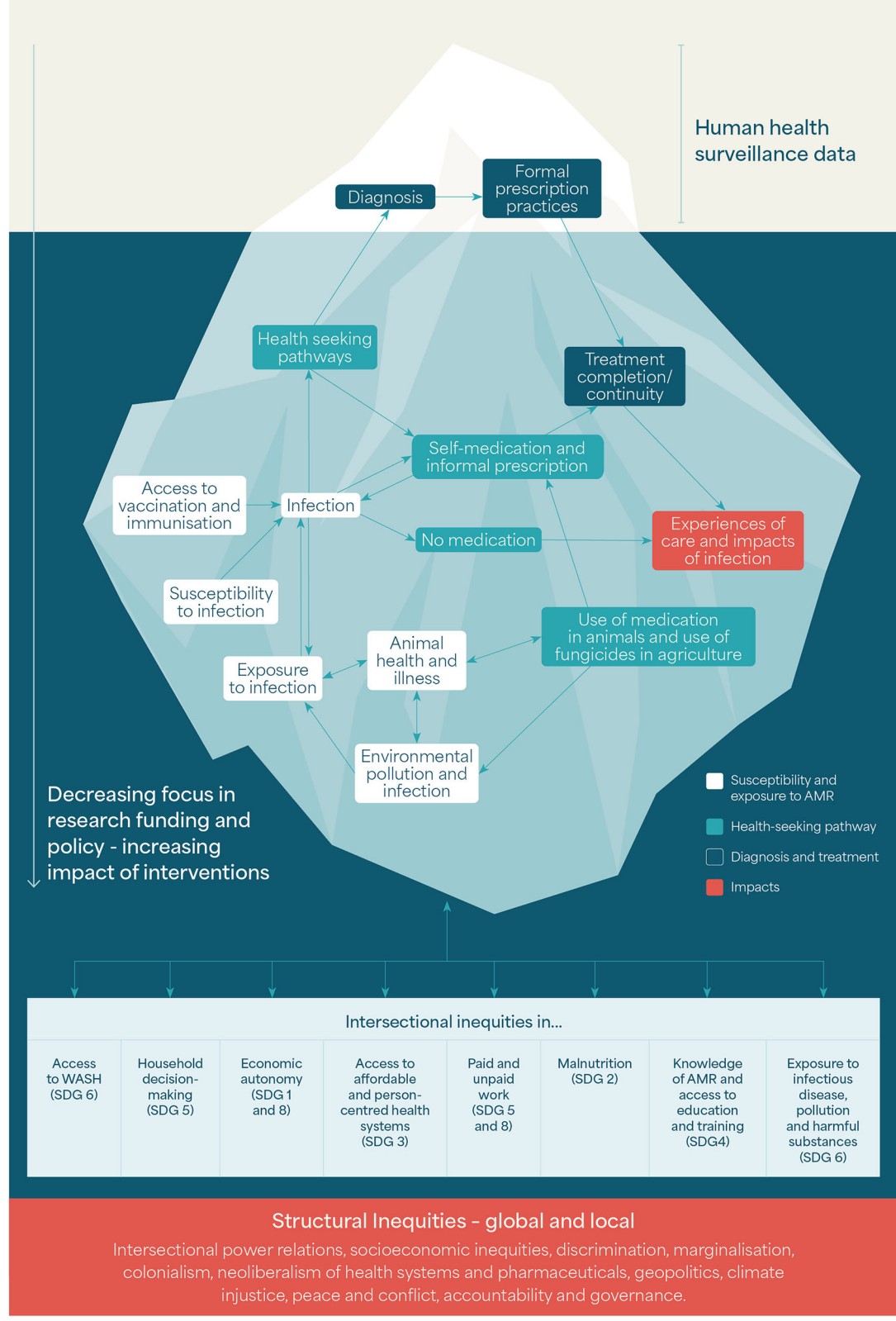

**Fig. 3 | Surveillance data as the tip of the iceberg: The structural root causes of AMR under the surface** (WASH water, sanitation and hygiene, SDG Sustainable Development Goal).

communities and contexts. These processes are aligned with what is understood about the social determinants of infectious disease and the intersectional nature of infectious diseases of poverty[17,132]. These processes relate to the ways in which power and resources are distributed in societies at multiple scales, from the global to the local, and

in often deep-rooted ways, and which ultimately cause diverse forms of harm to people living in these systems and power arrangements[44,133].

Our conceptual framework (Fig. 3) highlights how facility-collected surveillance data needs to be complemented with research into the structural drivers of AMR to support equitable evidence

production and policy making. It draws from wider bodies of literature on health inequities to show that these processes have roots that are both local and impossible to disentangle from vast global power inequities in health and politics more widely[134,135]. As Allel et al.[66] emphasise, factors beyond hospital settings affect emergence and dissemination of AMR but have "been largely overlooked by decisionmakers and researchers" (p7). Surveillance data captures those who access health facilities and diagnostics. Much less is known about complex community-level infection rates and health seeking processes. Similarly, surveillance of resistant infections may miss or obscure the lived experiences of AMR for people who do not make it to health facilities and the slow structural violence of the intersection of poverty and resistant infections[136].

Focus on the structural drivers of infection and antimicrobial use is key to designing effective One Health interventions. For example, directly addressing water pollution to prevent the dissemination of drug resistant infections has the power to reduce infections, associated medical costs and lost work hours[137]. Yet, the vast majority of articles we screened conceptualised knowledge and behaviours as both problem and solution, a finding echoed by others[29]. Calls for interventions that move beyond narrow behavioural approaches to address poverty, precarity, access to WASH, healthcare systems and living conditions come from across One Health domains[65,66,68,86,92,127,138,139]. For example, access to timely diagnosis and treatment with the most appropriate antimicrobials is important for preventing AMR.

Our review identifies significant evidence gaps across LMIC contexts, with little evidence on how AMR intersects with ethnicity, caste, Indigeneity, disability, refugee status and urban informality. There are broad inferences in the ways in which environmental pollution shapes AMR in informal settlements but very little research has explored this in depth[140]. Likewise, while many articles in our review refer to informal health services as part of the 'behavioural' challenge of AMR, there is a lack of research that shifts the lens to centre informal or pluralistic health systems as essential for understanding antimicrobial use[141]. Refugee and humanitarian contexts are known to be highly exposed to AMR[142], but there is very little evidence of specific experiences of AMR of refugees or from humanitarian and conflict contexts in LMICs[143]. The vulnerability, systemic exclusion and marginalisation of these communities results in antibiotic use as an everyday occurrence[94]. The voices and perspectives of those with direct experience are currently lacking. This necessitates shifting power in surveillance systems through considering qualitative research alongside quantitative and community-designed and led surveillance approaches.

Our review has also identified a noticeable gap in literature exploring the structural drivers of antifungal resistance. For example, some studies suggest that precarity among farmers leads to high levels of use of antifungals[144,145]. Reflecting the anthropocentric dominance, research on equity dimensions of animal and, particularly, plant and environmental health among those involved in and influenced by, these sectors was lacking. Filling this evidence gap will therefore require both a transformation of AMR surveillance systems and the prioritisation of grounded empirical research approaches drawing from the social sciences to understand complex and context-specific social process[30].

Our review process revealed a disconnect between the literature on AMR and the literature on inequity in infectious disease more broadly[146], though many of the themes that have emerged in our review as structural drivers of inequity in AMR are common to other areas of infectious disease research and the social determinants of health in LMICs. Given the global lack of contextually specific evidence about equity and AMR, we argue that the AMR field needs to draw from this wider body of evidence. This includes socioeconomic inequities, power relations and discrimination in health systems[27,147,148], historic and ongoing colonialism[134], neoliberalism of health systems and geopolitics[13,149] climate and environmental injustice[150] and conflict[47,151].

The authors' research experiences span multiple social science disciplines and perspectives, and collectively we find value in actively bringing these framings together to understand the equity and power relations in AMR and advance an intersectional analysis. For example, our review has highlighted that a lack of access to WASH is an inequity that often leads to increased exposure to AMR, and which is influenced by intersecting factors such as gender, age, ethnicity, location and occupation[29]. Therefore, the significant body of research on the wider inequities in access to WASH should be of value for understanding AMR inequities and designing interventions. We illustrate this wider lens in Fig. 4, and drew on these bodies of literature in developing the conceptual framework in Fig. 3.

Such literature can also help shift the biomedical framing of AMR towards a recognition of it as a "creeping disaster", that requires integrated cross-sectoral and cross-border action and can only be addressed at the root[19]. AMR also intersects with rapid-onset environmental hazards and conflict contexts in long-term ways that are under researched[152–156]. While the topic did not emerge significantly in the articles in our review, we also note the growing body of literature that describes and investigates the risk of AMR in conflict and humanitarian contexts[157]. Many of the structural drivers that we identify in our review are significant factors influencing this risk, including the breakdown of WASH infrastructure, the creation of conditions that lead to the spread of infection and development of AMR, such as overcrowding and environmental pollution, and the breakdown of health systems and delays in people's ability to seek health care[142,156,158–162].

In an era of polycrisis (a term that emphasises the increasing convergence of poverty, disasters, conflict and climate change), the lived experience of AMR will be unlikely to revolve around any single theme we have identified[163]. For example, refugees may be exposed to conflict, climate change, displacement, urban informality and infectious disease outbreaks occurring simultaneously, each of which has links to AMR[142]. Differences in infection, treatment duration, cost, accessibility and side effects each intersect to impact experience. Any one of these issues are deeply political, highlighting the need to centre AMR as an issue of equity and justice[164,165].

## Strengths and limitations

A limitation of our review is the inclusion of only literature in English, which may have brought in language biases. We acknowledge that the reliance on published literature may have excluded other forms of knowledge and evidence that we advocate for the importance of here. Within our multidisciplinary team and diverse perspectives with authors whose experiences span multiple contexts, we drew on rich experiences as well as literature in our conceptual considerations. We are aware that many of these structural power processes are longstanding but given that this review required synthesis across diverse and disparate global literature, the scope of our review needed to be manageable, and we limited our search to a five-year time frame. While our study focused specifically on literature that explicitly discussed AMR and antimicrobials, we recognise that there are significant bodies of literature of value to understanding equity and AMR that do not focus specifically on AMR (as we illustrate in Fig. 4). This reflects a wider limitation in much AMR research and policy – that is, a failure to conceptualise AMR inequities as embedded in wider health inequities and social determinants of health. We note a significant lack of articles in our review that explicitly focus on antifungal resistance, which reflects a wider gap in this area. Finally, it is worth emphasising that these structural processes are embedded within global structural power relations. This review did not focus on inequities between countries. Research has focused previously on countries' inequitable access to drugs[5]. Many of the long-term structural drivers of AMR are results of long histories of global power relations.

We highlight how AMR inequities are embedded in structural drivers that relate to the ways in which power and resources are

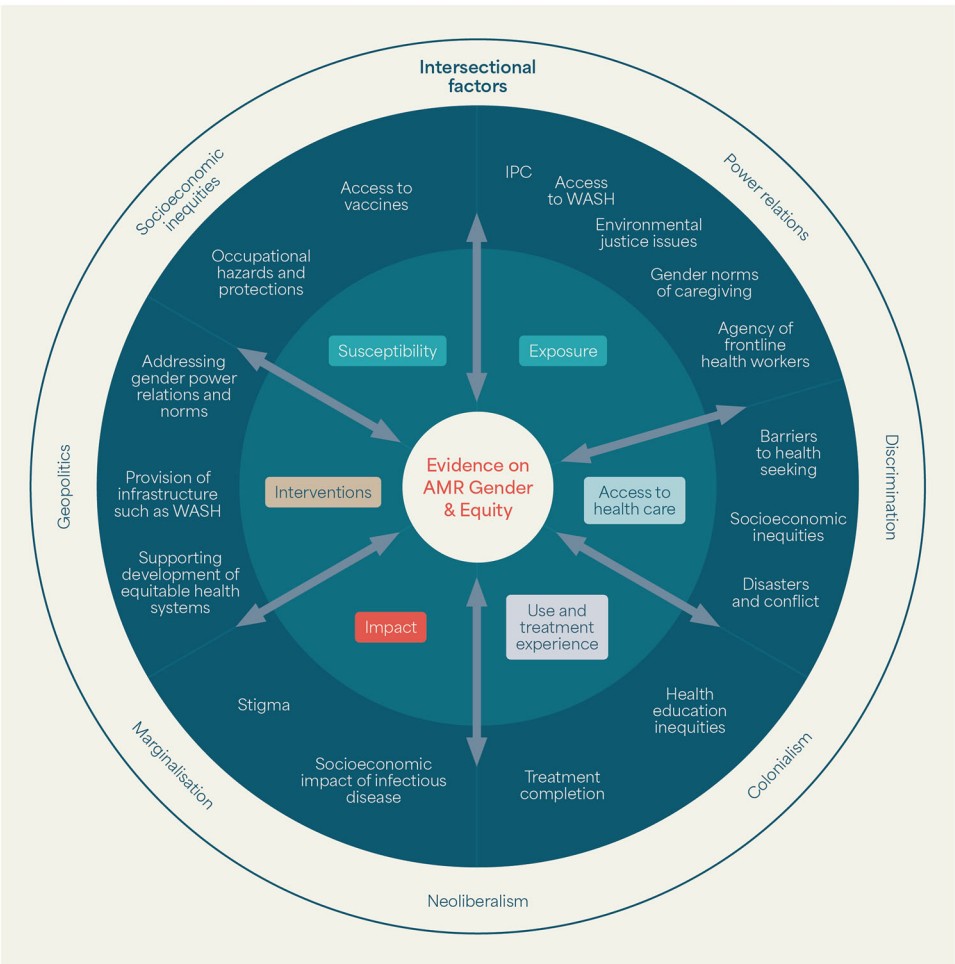

**Fig. 4 | Relevant bodies of evidence for understanding equity and AMR - highlighting the importance of an intersectional lens.** The inner green circle represents existing evidence that explores inequities relating to AMR. The outer green circle represents the wider bodies of evidence on inequities in health systems that are relevant to understanding inequities in AMR. The outermost circle emphasises that all these processes are embedded within intersectional, structural drivers on inequity. (WASH Water, sanitation and hygiene. IPC Infection prevention and control).

distributed in societies at multiple levels. Surveillance data currently reflects just the tip of the iceberg of the wider processes of AMR, and is limited by significant community level factors affecting health seeking. To illuminate the complex structural factors 'under the surface' of the iceberg, engagement with a much wider body of evidence on health inequities and the social determinants of health is required, alongside a shift away from harmful narratives of antibiotic 'misuse'. Future research should seek to address concerning evidence gaps around the underlying causes of inequities and injustices relating to AMR exposure, access to antimicrobials and lived experiences of AMR, with explicit attention to refugee and humanitarian contexts and those living in urban informal settlements. Attention to structural drivers requires multi-sectoral action and a focus on accountability, particularly for these populations particularly exposed or under-served by health systems. Research and action must also include the intersections of human health, animal health and environmental health, key dimensions of environmental and occupational justice and centre the voices of those most affected.

## Methods
### Critical Interpretive Synthesis
We undertook a critical interpretive synthesis (CIS) to examine the structural drivers of inequities in AMR with a focus on LMICs, using Dixon-Woods' CIS approach[166]. CIS is an approach to critically reviewing diverse, complex and sometimes disparate bodies of literature and evidence, both qualitative and quantitative, and its rigour does not depend on its systematic or reproducible nature, but rather on its thorough interpretive approach, grounded in methods of qualitative inquiry[50,166,167]. As Pahlman et al.[50] outline, CIS "aims not to simply aggregate and review findings or arguments, as is often the goal of systematic reviews, but rather to capture or 'take stock' of the key ideas, and offer reflections about the literature as a whole so as to inform further debate" (p2). It allows for the generation of a theory or conceptual framework with strong explanatory power, capturing key ideas and reflections from literature in order to contribute to nuanced debates as opposed to systematically reviewing every finding[24,167–170]. It is well suited for emergent and exploratory review questions. It is not intended to be a systematic review, but rather to take stock of key ideas and offer space for further discussion, and therefore appropriate to the complexity of applying an intersectional analysis to explore inequities in AMR[24,167].

### Search strategy
We developed a systematic search strategy with search terms (Table S1) based around three domains:

AMR domain: key terms relating to AMR and antibiotic use, access and stewardship.

Equity domain: key terms relating to equity, intersectionality and justice as well as terms focused on specific marginalised groups known to be impacted by AMR or infectious diseases more widely, drawing from overview literature[35,36].

LMIC domain: Names of all countries currently classified as low- or middle-income[171] and LMIC-related terms.

We searched three databases (Medline, Scopus and Global Health) for peer-reviewed studies examining the intersection of AMR and equity in LMICs, selected due to their collective comprehensiveness in covering the research topic. We conducted forward and backward citation searches from key literature to identify additional studies that may not have appeared in the initial database search, ensuring a more comprehensive and inclusive review of the literature. Searches were limited to 2019–2024 due to the large body of evidence and dynamic nature of AMR trends in public health.

#### Screening
Two independent reviewers separately carried out title and abstract screening of each article, with a third reviewer resolving any disputes. Screening was carried out using Rayyan online screening software and Microsoft Excel. To meet inclusion criteria, articles needed to describe findings related to how social or structural determinants of health impact on susceptibility or exposure to infection, transmission routes for AMR, access to treatment or the impact of the disease. Articles that reported only on burden of disease and did not explore reasons for identified trends were excluded. In addition, articles needed to focus on low- of middle-income countries[172], have been published within the last 10 years (2014–2024, in order to synthesise current evidence and provide up-to-date evidence summaries), and be peer-reviewed journal articles reporting on results of primary research or secondary data analysis (reviews and commentaries were excluded).

Full text screening was then carried out to assess articles' contribution to understanding of structural drivers of AMR. Papers were excluded if they only reported on the burden of infection among populations as these trends are reported elsewhere[4,8].

#### Data extraction and analysis
We used Microsoft Excel to create data extraction framework that was based initially on the WHO People-centred core package of AMR interventions[42], and was iteratively adapted as themes emerged during the process of familiarisation with and analysis of the literature into the following categories: susceptibility to infection, exposure to infection and antimicrobials, health-seeking pathways, self-medication and informal prescription, formal prescription practices, treatment completion/continuity, knowledge of AMR and antimicrobials, and experiences of care and impacts of infection. For each article, where relevant, quotations were extracted under each of these categories/themes (Table S3). We subsequently used qualitative content analysis to synthesise qualitative findings within and across themes.

As is common in CIS, the diverse nature of the types of studies included meant it was not appropriate to classify articles with standard quality assessment tools. CIS typically focus on inclusion according to contribution[173].

#### Reporting summary
Further information on research design is available in the Nature Portfolio Reporting Summary linked to this article.

## Data availability
Data included in this article is sourced from publicly available peer reviewed journal articles. We have included a list of included articles alongside our analysis decisions in the supplementary materials as Table S4: Included articles.

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

## Acknowledgements

This project is funded by the Department of Health and Social Care (DHSC)'s Fleming Fund using UK aid. The views expressed in this publication are those of the authors and not necessarily those of the UK DHSC or its Management Agent, Mott MacDonald. Grant number: FF152-558 (all authors).

## Author contributions

Conceptualisation and study design: K.D., R.S., S.T., and R.D., Methodology: K.D., R.S., Software: N.A. Validation: N.A.; Formal analysis: K.D., R.S.; Investigation: N.A.; Resources: N.A.; Data curation: K.D., R.L., B.E., M.M., R.S.; Writing – original draft: K.D., R.S.; Writing – review and editing: K.D., R.L., M.M., B.E., S.P., S.O., O.N., A.A., S.T.A., A.P., M.P.K., B.A., A.S.W.N., K.H., R.D., S.M.A., M.I., J.T., F.E.Y.A., W.M., L.O., S.R., S.B., M.G., S.T., R.S.; Visualisation: N.A.; Supervision: N.A.; Project administration: N.A.; Funding acquisition: S.T., R.D., R.S., L.O., M.G., W.M., S.R., S.B.; Data extraction: K.D., M.M., R.L., B.E., R.S.; All authors reviewed and inputted to drafts and approved the final version. Writing – original draft: K.D., R.S.; Writing – review and editing: K.D., R.L., M.M., B.E., S.P., S.O., O.N., A.A., S.T.A., A.P., M.P.K., B.A., A.S.W.N., K.H., R.D., S.M.A., M.I., J.T., F.E.Y.A., W.M., L.O., S.R., S.B., M.G., S.T., R.S.; Visualisation: N.A.; Supervision: N.A.; Project administration: N.A.; Funding acquisition: S.T., R.D., R.S., L.O., M.G., W.M., S.R., S.B.; Data extraction: K.D., M.M., R.L., B.E., R.S.; All authors reviewed and inputted to drafts and approved the final version.

## Competing interests

The authors declare no competing interests.
