## [Transparent Peer Review file · Nature Communications]

Antimicrobial resistance, equity and justice in low- and middle-income countries: an intersectional critical interpretive synthesis

Corresponding Author: Dr Katy Davis

Version 0:

Reviewer comments:

Reviewer #1

(Remarks to the Author)

In this review, the Authors perform a critical interpretive synthesis investigating the structural drivers of antimicrobial resistance in low and middle-income countries and describe solutions to remedy these drivers. I found this review very interesting and timely. I have mostly minor suggestions.

1. Line 49: Can you include a few other targets regarding AMR-related inequities here beyond improving access to safe housing and clean water?
2. Line 56: What does the "(1677)" refer to here? Page number? If so I think it can be removed.
3. Line 57: I think you mean "environment" rather than "environmental" here.
4. Line 105: Can you please define "WASH" here?
5. In Figure 3 you note "Use of medication in animals" as a driver of AMR. This is certainly true. I would also consider adding use of fungicides in agriculture, if this is applicable to LMIC's. This is a big driver in antifungal resistance in fungi.
6. Line 152: "Meloidosis" does not need to be capitalized.
7. Line 155: "Malaria" does not need to be capitalized.
8. Line 158: Can you describe which drug-resistant infections sex workers were at risk for in this study?
9. In the "Exposure to infections and antimicrobials" section, did you come across any papers that describe the use of fungicides in agriculture and antifungal resistance in fungi in LMIC's?
10. Line 195-196: I think you mean "...or self-medication for malaria, for example, can lead to resistance."
11. In the "Access to diagnosis and healthcare", did you find any mention of political instability leading to difficulty accessing healthcare in your research? For example, when the Soviet Union crumbled so did their centralized health system and a lot of individuals being treated for tuberculosis at the time were unable to access medication. This led to increases in drug-resistant TB in Russia and other countries in the former Soviet Union. Wondering if this has occurred elsewhere following political upheaval? In addition, conflict, such as has been seen in Sudan and Gaza, has severely impacted healthcare infrastructure.
12. In the "Self-medication with antimicrobials" section, can you discuss that medicines (including antibiotics) are often able to be purchased without a prescription in LMIC's due to poor healthcare infrastructure.
13. Line 300: I think you mean "This is echoed..."
14. Line 303: "women living with..." what? Do you mean DR-TB?
15. Line 306: I think you mean "as reported by HIV-positive..."
16. Line 328: Does the page number need to be included here?
17. Line 333: This citation is not uniform with the rest.
18. Line 334: Can you please define "WHO GLASS"?
19. Line 351: Similarly, is the page number necessary here?
20. Line 354: "refugee status, and urban informality".
21. Line 358: I think there is a typo in "centre informal pr pluralistic health..."
22. Line 371: I don't think you need a comma here.
23. Line 454: In this section "Search Strategy", I would refer to Table 1 here which describes your search terms.
24. Line 478 - 480: I think you mean this to be one sentence. "...use of qualitative and quantitative data it was not appropriate..."

Reviewer #2

(Remarks to the Author)

Noteworthy Results

- Figure 3 presents a compelling conceptual framework that illustrates the socio-cultural components contributing to antimicrobial resistance (AMR). However, the written results section does not adequately address or connect the specific components depicted in Figure 3.

Significance to the Field

- The work appears to be significant as it addresses the burden of AMR, particularly in low- and middle-income countries (LMICs). It adds to the existing literature by highlighting the socio-cultural aspects of AMR while presenting a novel framework.

Support for Conclusions

- The work supports its conclusions to some extent but requires additional evidence, particularly in the results section where connections to Figure 3 are not clearly articulated. The discussion introduces concepts not previously addressed in the results, necessitating further development and citation.

Flaws in Data Analysis and Interpretation

- The results section fails to reference or explain Figure 3 adequately, which limits understanding. Additionally, some sentences in the discussion are unclear due to awkward phrasing, which could hinder comprehension. These issues may require revision but do not necessarily prohibit publication.

Methodology Soundness

- The methodology appears sound, particularly in its reliance on surveillance data. However, the lack of contextually specific understanding of AMR drivers raises questions about the completeness of the methodology. It meets expected standards but could benefit from more rigorous detail.

Reproducibility of Methods

- There is insufficient detail provided in the methods section for the work to be easily reproduced. More comprehensive descriptions of the data sources and analytical techniques are needed to ensure reproducibility.

Additional Suggestions

- **Rewording and Clarity:** Several suggestions for rewording sentences to enhance clarity have been made, particularly regarding the use of informal language and awkward phrasing.
- **Structuring the Results:** It is recommended to restructure the results section to align with the components of Figure 3, possibly by introducing subheadings that correspond to each component.
- **Citations and Context:** The discussion should include citations for concepts not addressed in the results, ensuring that all claims are well-supported and contextualized within the authors' findings.

Introduction

The following sentence seems to be missing something:

Line 24: AMR therefore disproportionately impacts LMICs, where up to 90 percent of deaths from AMR occur.

The following sentence seems informal, the use of the word "huge" seems out of place for this journal:

Line 21: How antimicrobials are used has a huge impact on the risk of resistance developing.

Use of "on how because" is awkward and confusing, I suggest rewording:

Line 28: Within countries, AMR burden is also unequally distributed (9,10), but we do not have detailed data on how because there is an absence of analysis of disaggregated AMR data at a global level.

Results

Figure 3 is fantastic, it adds an compelling conceptual framework to the existing literature in terms of socio-cultural components contributing to AMR. Nevertheless, the written section of the results does not seem to address the specific components presented in Figure 3. Specifically, the results do not describe the connections made in the figure but rather address specific components of AMR without tying them together in a succinct way as is done in Figure 3. Results do not reference Figure 3 (lines 114-307), thus it is difficult to comprehend the connection with the literature review and findings in regard to development of the figure. Maybe one way to improve this would be to ensure that each component of Figure 3 has a subheading in the results with a paragraph connecting all of the conceptual components. Again, I suggest restructuring the results to facilitate and describe development of Figure 3, which is the most compelling part of the manuscript. Another potential option is to draw on the concepts highlighted in the first sentences of each paragraph in the discussion to organize presentation of the results this way.

Discussion:

In the first paragraph of the discussion there is mention of concepts that have not been addressed in the results section. Thus, the sentence needs a citation to support the components, and they also need to be developed and explained in the context of the authors' findings.

Line 322 :These include poverty, marginalisation of groups and individuals, colonialism, neoliberalism in health systems, geopolitics, climate justice, conflict and governance.

The paragraph on reliance on surveillance data is really interesting and is one of the major strengths of the paper, especially the following components:

Line 335: An implication of this is that we do not currently know who the burden of AMR is affecting most, either by gender or age group, and we have little contextually-specific understanding of the social processes shaping AMR drivers. Currently, therefore, surveillance data can only provide a limited understanding to support equitable evidence production and policy making.

The following sentence seems to have an additional word making it difficult to understand:

Line 369: Our review simultaneously highlights a disconnect between AMR literature and that on health inequity relating to infectious disease (117).

The same is true for the following sentence:

Line 372: Given the evidence gaps in contextually specific understandings of equity and AMR in many geographies and focus populations, we suggest widening the bounds of evidence considered relevant to AMR.

Use of the term "creeping disaster" in the last paragraph of the discussion is awkward. It seems as if it is being used to define a new term in the context of AMR but is not defined in the results as a finding.

Reviewer #3

(Remarks to the Author)

This study describes findings from a critical review meant to illuminate structural drivers of inequities in AMR in LMICs. The authors are correct that this perspective is lacking in current approaches to mitigate AMR and I appreciated their discussion of the fact that healthcare based surveillance is unlikely to reveal the root causes of AMR.

However, I found the methods section to be exceptionally vague, which limits the reproducibility of this work. I also would appreciate more details about their motivation for using intersectional approaches. Specifically, the significance of this approach for the field remains unclear.

Major comments:

The methods section lacks multiple key details that are critical for a review of the literature.

1) The search strategy mentions "key terms." It should be clarified that these key terms are available in the appendix.

2) What is "blinded" title and abstract screening? Reviewers were blinded to the author list?

3) Inclusion and exclusion criteria must be explicitly listed. All that is written right now is "articles needed to report on empirical research explicitly addressing inequities relevant to AMR in LMIC contexts" What do the authors mean by empirical research? "explicitly addressing inequities relevant to AMR in LMIC contexts" is subjective and not reproducible.

4) I couldn't follow most of the Data Extraction and Analysis section. What do the authors mean when they say "categories" for data extraction and analysis were developed "inductively" and "deductively"? What data were actually extracted from each article?

The authors say that "quality of the research does not affect the relevance of their findings to our research question." This is a strong statement. Can the authors explain why this might be the case?

The authors say they assessed papers for "conceptual relevance." What does this mean?

Papers were categorized into three categories. For what purpose, extraction, analysis, or another purpose? I don't see these categories mentioned again in the Findings. Did two reviewers do this classification? It seems this classification could be very subjective.

5) Was this review registered on Prospero or elsewhere?

In the Introduction, the authors state intersectional approaches are needed but are fairly new in the AMR field. They say this represents a missed opportunity to better tailor AMR interventions. It is important that the author share concrete examples of how an intersectional approach has informed interventions to improve programmatic effectiveness, equity, and sustainability for other disease outcomes. Otherwise, this review could be construed as more of a thought exercise rather than a paradigm shift.

Minor comments:

1) Some sentences could use more specificity:

Lines 157-158: What resistant infections?

Line 164: What mining chemicals? Susceptibility to TB or infection more broadly?

Line 193-194: Sentence about vector borne disease, what does this have to do with AMR, specifically?

Line 271: Shukla et al - is this in Tanzania, too? High level relative to urban dwellers?

line 230: What was found in China?

Line 304: Redwood et al paper - among whom?

2) A scoping review on the intersections between AMR pathogens and the SDOH was recently published in the Lancet Microbe. This would be relevant to include in introduction and/or discussion. PMID: 39653050.

3) Line 354: The authors say their review highlights how AMR intersects with caste, disability, refugee status... I see no mention of caste or disability status in their results. I see 1 reference to prescribers in refugee campus, not individuals' refugee status.

4) There were numerous typos and mis-formatted sentences and citations throughout the manuscript:

Line 152, 155: Why are diseases capitalized?

Line 195-196: Shouldn't be a period before malaria

Line 300: "This is echoes..." Sentence has poor grammar, also what findings?

Line 306: as reported "among" HIV-positive...

Line 333: Doron et al citation is not in correct format

Line 358: informal "or" pluralistic

Lines 478-480 are two sentences but should be one sentence

Reviewer #4

(Remarks to the Author)

The manuscript addresses relevant concerns related to the study of equity and antimicrobial resistance (AMR) through a critical interpretive synthesis (CIS) approach. While these are timely concerns, many of the issues discussed under the identified themes have already been discussed in previous CIS (see Pahlman et al., 2022, and Aguiar et al., 2024). A clearer positioning of the current manuscript in relation to existing CIS literature addressing AMR concerns would allow the authors to enrich the discussion and would enhance the added value of this manuscript.

Even considering the readership and approach of Nature Communications, for a CIS engaging with critical theory, the article needs to be more rigorous in its approach to equity, (environmental) justice and intersectionality. In presenting a conceptual framework that is based on a CIS informed by critical theory, the manuscript would benefit from stronger critical grounding to inform a more engaged analysis of the structural inequities as they relate to AMR. A stronger engagement of intersectional theory across the relevant sections of the manuscript would also enable a deeper and theory-driven analysis of the structural inequities the authors themselves identify as key evidence gaps.

The global dimensions of structural inequities are highlighted several times throughout the manuscript. This suggests that the authors recognize their importance for the conceptual framework and manuscript, however its importance is only briefly mentioned in the limitations section.

The article offers a valuable synthesis of the evidence on structural inequities as they relate to AMR, but it would benefit from grounding the analysis of equity and (environmental) justice in conceptualizations aligned with critical or intersectional theory, a deeper and more rigorous engagement with intersectional frameworks, a relevant positioning within existing CIS literature, and a more substantive exploration of the global dimensions of AMR inequities, to reach its full potential.

Version 1:

Reviewer comments:

Reviewer #2

(Remarks to the Author)

Thank you for the opportunity to review changes to the manuscript. Concerns expressed in the review comments I provided have been addressed in the revisions. Thank you

Reviewer #3

(Remarks to the Author)

The authors have adequately responded to most of my comments. Some points remain unclear; for example, I asked: "What data were actually extracted from each article?" The authors have shared that they used a data extraction framework that was "iteratively adapted" into several categories. It is still not clear what data were actually extracted from each article, regardless of the category that the article was binned into. Including a data extraction template as an appendix would be helpful.

I continue to have concerns with the subjective nature of this research approach. In response to my previously stated concerns, the authors wrote: "the rigour does not depend on its systematic or reproducible nature, but rather on its thorough interpretive approach and [that it is?] grounded in methods of qualitative inquiry." The authors have also clarified that articles could meet inclusion criteria even if they merely speculated on drivers of AMR inequities in their discussion, rather than rigorously investigated them. So, not only is the authors approach not meant to be reproducible, but the speculations of others were considered valid inputs to their synthesis of the literature..? Overall, I question how this approach might genuinely move the field forward or inform further debate as the authors have stated is their purpose.

Reviewer #4

(Remarks to the Author)

The authors have effectively addressed all comments and concerns, and I believe the revised version will make a significant contribution to AMR research.

Reviewer	Section	Comment	Response to reviewer
1		Line 49: Can you include a few other targets regarding AMR-related inequities here beyond improving access to safe housing and clean water?	We have added to this sentence and this section now reads: "(20). We aim to counteract this narrative by highlighting what is just as necessary is to address the root causes of AMR related inequities. This might include improving access to safe housing and water (20), extending health coverage to rural and informal communities or addressing economic inequities and inequitable gender norms."
		Line 56: What does the "(1677)" refer to here? Page number? If so I think it can be removed.	We have added "p" to show that this is a page number for a quote. Happy to remove if that meets journal editorial policy.
		Line 57: I think you mean "environment" rather than "environmental" here.	Edit made
		Line 105: Can you please define "WASH" here?	Definition added
		In Figure 3 you note "Use of medication in animals" as a driver of AMR. This is certainly true. I would also consider adding use of fungicides in agriculture, if this is applicable to LMIC's. This is a big driver in antifungal resistance in fungi.	
		Line 152: "Melioidosis" does not need to be capitalized.	Edit made
		Line 155: "Malaria" does not need to be capitalized.	Edit made
		Line 158: Can you describe which drug-resistant infections sex workers were at risk for in this study?	We have added metnion of the specific infections that the literature describes

		In the "Exposure to infections and antimicrobials" section, did you come across any papers that describe the use of fungicides in agriculture and antifungal resistance in fungi in LMIC's?	Our review did not identify any studies that explicitly explored the structural drivers of antifungal resistance. This is a noticeable gap and we have added a note about this to the discussion and also to the limitations. We have also mentioned this important gap in the introduction and referenced a key review on antifungal resistance.
		Line 195-196: I think you mean "...or self-medication for malaria, for example, can lead to resistance."	Edit made
		In the "Access to diagnosis and healthcare", did you find any mention of political instability leading to difficulty accessing healthcare in your research? For example, when the Soviet Union crumbled so did their centralized health system and a lot of individuals being treated for tuberculosis at the time were unable to access medication. This led to increases in drug-resistant TB in Russia and other countries in the former Soviet Union. Wondering if this has occurred elsewhere following political upheaval? In addition, conflict, such as has been seen in Sudan and Gaza, has severely impacted healthcare infrastructure.	This is a really important theme and unfortunately did not emerge as a significant theme in the articles in our review. We have added reference to the growing body of literature on this topic, and particularly in Gaza and Ukraine, to the introduction and discussion.
		In the "Self-medication with antimicrobials" section, can you discuss that medicines (including antibiotics) are often able to be purchased without a prescription in LMIC's due to poor healthcare infrastructure.	We have added a sentence under this section that now reads: "'Self-medication' refers to the seeking of antimicrobials without prescription, either through informal providers, family or social networks, including in contexts where antibiotics are available to purchase without a prescription"

	Line 300: I think you mean "This is echoed..."	Edit made
	Line 303: "women living with..." what? Do you mean DR-TB?	Edit made
	Line 306: I think you mean "as reported by HIV-positive..."	Edit made
	Line 328: Does the page number need to be included here?	We have removed page number
	Line 333: This citation is not uniform with the rest.	We have updated this citation
	Line 334: Can you please define "WHO GLASS"?	We have added full name here
	Line 351: Similarly, is the page number necessary here?	We would normally include a page number for an extended quote like this but are happy to defer to journal editorial policy here
	Line 354: "refugee status, and urban informality".	Edit made
	Line 358: I think there is a typo in "centre informal pr pluralistic health..."	Edit made
	Line 371: I don't think you need a comma here.	Edit made
	Line 454: In this section "Search Strategy", I would refer to Table 1 here which describes your search terms.	Added reference to search terms table in methods section
	Line 478 - 480: I think you mean this to be once sentence. "...use of qualitative and quantitative data it was not appropriate..."	Edit made

2		Noteworthy results: Figure 3 presents a compelling conceptual framework that illustrates the socio-cultural components contributing to antimicrobial resistance (AMR). However, the written results section does not adequately address or connect the specific components depicted in Figure 3.	We have updated the sub headings in the results and the sections of the flow chart in the iceberg figure so that they are consistent and link more directly. We have also added sections under each results theme to highlight the links between these themes in line with the framework in figure 3. We have also added a brief summary of results paragraph that illustrates links between dimensions of the framework in figure 3. We have also added detail to the description of the framework in the results section.
		Significance to the field: The work appears to be significant as it addresses the burden of AMR, particularly in low- and middle-income countries (LMICs). It adds to the existing literature by highlighting the socio-cultural aspects of AMR while presenting a novel framework.	
		Support for Conclusions: The work supports its conclusions to some extent but requires additional evidence, particularly in the results section where connections to Figure 3 are not clearly articulated. The discussion introduces concepts not previously addressed in the results, necessitating further development and citation.	Thanks for these comments. To address both, we have updated the sub headings in the results and the sections of the flow chart in the iceberg figure so that they are consistent and link more directly. We have also added sections under each results theme to highlight the links between these themes in line with the framework in figure 3. We have also added a brief summary of results paragraph that illustrates links between dimensions of the framework in figure 3. We have also added detail to the description of the framework in the results section.
		Flaws in Data Analysis and Interpretation: The results section fails to reference or explain Figure 3 adequately, which limits understanding. Additionally, some sentences in the discussion are unclear due to awkward phrasing, which could hinder comprehension. These issues may require	We have also edited and clarified sections that you highlight in the discussion

		revision but do not necessarily prohibit publication.	
		Methodology Soundness: The methodology appears sound, particularly in its reliance on surveillance data. However, the lack of contextually specific understanding of AMR drivers raises questions about the completeness of the methodology. It meets expected standards but could benefit from more rigorous detail.	The lack of contextually specific AMR drivers identified in this review is likely a reflection of the research gap. There is a significant gap in research that explicitly investigates the structural drivers of AMR, with the vast body of research on AMR and inequity describing inequitable trends in the burden of AMR but not exploring the social root causes of these inequities. Searching for literature that explores the structural drivers of AMR is a complex challenge due primarily to the diversity of language used to describe complex, social/structural drivers and inequities and we have developed detailed and expansive search terms and strategies that draw from qualitative enquiry and interpretive synthesis to maximise our ability to identify relevant literature (see Dixon-Woods et al 2006, referenced in the manuscript)

		Reproducibility of Methods: There is insufficient detail provided in the methods section for the work to be easily reproduced. More comprehensive descriptions of the data sources and analytical techniques are needed to ensure reproducibility.	Thanks for this feedback. We have added detail on methods in both the introduction and the methods section. We have provided more detail on the CIS approach and its approach to synthesis and rigour that draw from qualitative inquiry. We have added references to other CIS reviews in the field of AMR and detailed how our review builds on their findings. This section now reads: "CIS is an approach to critically reviewing diverse, complex and sometimes disparate bodies of literature and evidence, both qualitative and quantitative, and its rigour does not depend on its systematic or reproducible nature, but rather on its thorough interpretive approach and grounded in methods of qualitative inquiry (35,163,164). As Pahlman et al (35) outline, CIS "aims not to simply aggregate and review findings or arguments, as is often the goal of systematic reviews, but rather to capture or 'take stock' of the key ideas, and offer reflections about the literature as a whole so as to inform further debate" (p2). It allows for the generation of a theory or conceptual framework with strong explanatory power, capturing key ideas and reflections from literature in order to contribute to nuanced debates as opposed to systematically reviewing every finding (36,164–167). It is well suited for emergent and exploratory review questions. It is not intended to be a systematic review, but rather to take stock of key ideas and offer space for further discussion and therefore appropriate to the complexity of applying an intersectional analysis to explore inequities in AMR."
--	--	---	---

	Additional Suggestions: Rewording and Clarity: Several suggestions for rewording sentences to enhance clarity have been made, particularly regarding the use of informal language and awkward phrasing.	Many thanks. We have made the suggested edits throughout. See other comments for details.
	Structuring the Results: It is recommended to restructure the results section to align with the components of Figure 3, possibly by introducing subheadings that correspond to each component.	We have updated the sub headings in the results and the sections of the flow chart in the iceberg figure so that they are consistent and link more directly. We have also added sections under each results theme to highlight the links between these themes in line with the framework in figure 3. We have also added a brief summary of results paragraph that illustrates links between dimensions of the framework in figure 3. We have also added detail to the description of the framework in the results section.
	Citations and Context: The discussion should include citations for concepts not addressed in the results, ensuring that all claims are well-supported and contextualized within the authors' findings.	We have more explicitly brought out these themes in the text describing the conceptual framework and its development in the results section. We reference and describe these as we introduce them. We have added definitions for terms new to the discussion such as 'polycrisis'. Additionally, we have added a definition of "creeping disaster" to the introduction.
Introduction	The following sentence seems to be missing something: Line 24: AMR therefore disproportionately impacts LMICs, where up to 90 percent of deaths from AMR occur.	we have reworded this section to read: " Low- and Middle-Income Countries (LMICs) experience high rates of infectious disease, challenges in access to healthcare and global inequities relating to supply of antimicrobial access (4,5), and up to 90 percent of total global deaths from AMR (6). AMR, therefore, disproportionately impacts LMICs. "

	Introduction	The following sentence seems informal, the use of the word “huge” seems out of place for this journal: Line 21: How antimicrobials are used has a huge impact on the risk of resistance developing.	We have edited this to 'significant'
	Introduction	Use of “on how because” is awkward and confusing, I suggest rewording: Line 28: Within countries, AMR burden is also unequally distributed (9,10), but we do 29 not have detailed data on how because there is an absence of analysis of disaggregated AMR 30 data at a global level.	We have reworded this sentence to read: " Within countries, AMR burden is also unequally distributed (9,10), but we do not have a detailed understanding of this distribution because there is an absence of analysis of disaggregated AMR data at a global level. "

	Results	Figure 3 is fantastic, it adds an compelling conceptual frawork to the exisitng literature in terms of socio-cultural components contributing to AMR. Nevertheless, the written section of the results does not seem to address the specific components presented in Figure 3. Specifically, the results do not describe the connections made in the figure but rather address specific components of AMR without tying them together in a succinct way as is done in Figure 3. Results do not reference Figure 3 (lines 114-307), thus it is difficult to comprehend the connection with the literature review and findings in regard to development of the figure. Maybe one way to improve this would be to ensure that each component of Figure 3 has a subheading in the results with a paragraph connecting all of the conceptual components. Again, I suggest restructuring the results to facilitate and describe development of Figure 3, which is the most compelling part of the manuscript. Another potential option is to draw on the concepts highlighted in the first sentences of each paragraph in the discussion to organize presentation of the results this way.	Thanks for this important feedback. We have added a summary of results paragraph that illustrates the processes and links identified between results and how these have been used to develop the links between the dimensions of the framework in figure 3. We have edited the sub headings so that those used in the resultst and the conceptual framework are consisten. Finally, we have added a paragraph in the results section (under the subheading 'conceptual framework') to clarify how the conceptual framework draws directly on the results themes.
--	---------	---	---

	Discussion	In the first paragraph of the discussion there is mention of concepts that have not been addressed in the results section. Thus, the sentence needs a citation to support the components, and they also need to be developed and explained in the context of the authors' findings. Line 322 :These include poverty, marginalisation of groups and individuals, colonialism, neoliberalism in health systems, geopolitics, climate justice, conflict and governance.	We have more explicitly brought out a number of the themes from the framework in the text describing the conceptual framework and its development in the results section, where we include citations for these. We have added definitions and citations for terms new to the discussion such as 'polycrisis'. Additionally, we have added a definition of "creeping disaster" to the introduction.
	Discussion	The paragraph on reliance on surveillance data is really interesting and is one of the major strengths of the paper, especially the following components: Line 335: An implication of this is that we do not currently know who the burden of AMR is affecting most, either by gender or age group, and we have little contextually-specific understanding of the social processes shaping AMR drivers. Currently, therefore, surveillance data can only provide a limited understanding to support equitable evidence production and policy making.	Thank you - this is one of the key factors that drove the development of our framework.
	Discussion	The following sentence seems to have an additional word making it difficult to understand: Line 369: Our review simultaneously highlights a disconnect between AMR	We have reworded this sentence to read: "Our review simultaneously highlights a disconnect between the literature on AMR and the literature on inequity in infectious disease more broadly."

		literature and that on health inequity relating to infectious disease (117).	
Discussion		The same is true for the following sentence: Line 372: Given the evidence gaps in contextually specific understandings of equity and AMR in many geographies and focus populations, we suggest widening the bounds of evidence considered relevant to AMR.	We have reworded this sentence for clarity. It now reads: "Given the global lack of contextually specific evidence about equity and AMR, we argue that the AMR field needs to draw from this wider body of evidence on health inequity and social determinants of health to understand equity dimensions of AMR."
Discussion		Use of the term "creeping disaster" in the last paragraph of the discussion is awkward. It seems as if it is being used to define a new term in the context of AMR but is not defined in the results as a finding.	We have edited a line in the introduction where we introduce the concept of a 'creeping disaster', and suggest that AMR is an example of a creeping disaster. This sentence now reads: " The drivers of AMR therefore constitute a 'creeping disaster' – a term increasingly used to describe complex, deep rooted and inequitable process that lack definable temporal and spatial boundaries". In the discussion, we have added to the sentence that you highlight and this now reads: "Drawing on diverse bodies of literature can also help shift the biomedical framing of AMR towards a recognition of it as a "creeping disaster", that requires integrated cross-sectoral and cross-border action and can only be addressed at the root (13). Creeping disasters (unlike highly-visible, rapid-onset disaster events) can go under the radar of major discourses and the responses of actors and policymakers in humanitarian and disaster governance (121,122)."

3	General	This study describes findings from a critical review meant to illuminate structural drivers of inequities in AMR in LMICs. The authors are correct that this perspective is lacking in current approaches to mitigate AMR and I appreciated their discussion of the fact that healthcare based surveillance is unlikely to reveal the root causes of AMR. However, I found the methods section to be exceptionally vague, which limits the reproducibility of this work. I also would appreciate more details about their motivation for using intersectional approaches. Specifically, the significance of this approach for the field remains unclear.	Thanks for this feedback. We have added detail on methods in both the introduction and the methods section. We have provided more detail on the CIS approach and its approach to synthesis and rigour that draw from qualitative inquiry. We have added references to other CIS reviews in the field of AMR and detailed how our review builds on their findings. The methods section now reads: "CIS is an approach to critically reviewing diverse, complex and sometimes disparate bodies of literature and evidence, both qualitative and quantitative, and its rigour does not depend on its systematic or reproducible nature, but rather on its thorough interpretive approach and grounded in methods of qualitative inquiry (1–3). As Pahlman et al (1) outline, CIS “aims not to simply aggregate and review findings or arguments, as is often the goal of systematic reviews, but rather to capture or ‘take stock’ of the key ideas, and offer reflections about the literature as a whole so as to inform further debate” (p2). It allows for the generation of a theory or conceptual framework with strong explanatory power, capturing key ideas and reflections from literature in order to contribute to nuanced debates as opposed to systematically reviewing every finding (1–5). It is well suited for emergent and exploratory review questions. It is not intended to be a systematic review, but rather to take stock of key ideas and offer space for further discussion and therefore appropriate to the complexity of applying an intersectional analysis to explore inequities in AMR."
---	---------	---	---

			We have added more detail to the inclusion criteria in the main text under 'screening' subheading and the appendix, and, building on comments about the use of the term 'conceptual relevance') have replaced this with more detailed description of the types of study findings that we included in this review. Under the 'data extraction' subheading, we have added more detail on the coding framework. This now reads: "We used data extraction framework that was based initially on the WHO People-centred core package of AMR interventions (33), and was iteratively adapted as themes emerged during the process of familiarisation with and analysis of the literature into the following categories: susceptibility to infection, exposure to infection and antimicrobials, health-seeking pathways, self-medication and informal prescription, formal prescription practices, treatment completion/continuity, knowledge of AMR and antimicrobials, and experiences of care and impacts of infection." With regards to motivation for use of intersectional approaches: we have added more detail and reworded paragraphs in the introduction between pages 2 and 4, including more clarity on the value of intersectional lens to AMR and examples of value of use of intersectional lens in health more widely.
--	--	--	---

	Methods	The search strategy mentions "key terms." It should be clarified that these key terms are available in the appendix.	We have added the following to the search strategy section of the methods section: "We developed a systematic search strategy with search terms (Table S1) based around three domains"
	Methods	What is "blinded" title and abstract screening? Reviewers were blinded to the author list?	Thank you for highlighting this - we meant independent so we have edited this wording to read: "Two independent reviewers separately carried out title and abstract screening of each article, with a third reviewer resolving any disputes"
	Methods	Inclusion and exclusion criteria must be explicitly listed. All that is written right now is "articles needed to report on empirical research explicitly addressing inequities relevant to AMR in LMIC contexts" What do the authors mean by empirical research? "explicitly addressing inequities relevant to AMR in LMIC contexts" is subjective and not reproducible.	We have included a table of inclusion and exclusion criteria in the appendix. In this table, as well as in the methods text, we have added more detailed description of the inclusion criteria that you mention. This specific criteria now reads: Article must... "Provide insights into how social or structural determinants of health impact on susceptibility or exposure to infection, transmission routes for AMR, access to treatment, or the impact of the disease through either: - directly exploring equity dimensions of, and drivers of inequity in, AMR; - reflecting in the discussion about what might underly AMR inequities; - reporting incidental findings on equity dimensions of AMR"

	Methods	I couldn't follow most of the Data Extraction and Analysis section. What do the authors mean when they say "categories" for data extraction and analysis were developed "inductively" and "deductively"? What data were actually extracted from each article?	We have edited this section for clarity and have included a list of the themes in the coding framework. We have clarified what we had meant by 'inductive' and 'deductive' and explained that the coding framework was iteratively developed - this section now reads: "We used data extraction framework that was based initially on the WHO People-centred core package of AMR interventions (1), and was iteratively adapted as themes emerged during the process of familiarisation with and analysis of the literature into the following categories: susceptibility to infection, exposure to infection and antimicrobials, health-seeking pathways, self-medication and informal prescription, formal prescription practices, treatment completion/continuity, knowledge of AMR and antimicrobials, and experiences of care and impacts of infection."
	Methods	The authors say that "quality of the research does not affect the relevance of their findings to our research question." This is a strong statement. Can the authors explain why this might be the case?	Thanks for highlighting that this was not clear and we have edited this section for overall clarity. In our approach, we were interested in any paper that reported on drivers and or explored issues of power structures, which is what we originally meant by conceptually relevant. However, we have removed due to lack of clarity and add instead this more specific inclusion criteria. We have included a sentence in the data extraction and analysis section: "As is common in CIS, the diverse nature of the types of studies included meant it was not appropriate to classify articles with standard quality assessment tools. "

	Methods	The authors say they assessed papers for "conceptual relevance." What does this mean?	We have removed this term as we agree it was unclear. We have added a more specific description: "To be included in our review, articles needed to provide insights into how social or structural determinants of health impact on susceptibility or exposure to infection, transmission routes for AMR, or the impact of the disease."
	Methods	Papers were categorized into three categories. For what purpose, extraction, analysis, or another purpose? I don't see these categories mentioned again in the Findings. Did two reviewers do this classification? It seems this classification could be very subjective.	We have reworded this section to clarify this, as we agree this was vague. These categories were used as a process to determine whether an article met inclusion criteria. This section now reads: "To be included in our review, articles needed to provide insights into how social or structural determinants of health impact on susceptibility or exposure to infection, transmission routes for AMR, or the impact of the disease. To do so, they needed to do one of the following: directly explore equity dimensions of, and drivers of inequity in, AMR; reflect in the discussion about what might underly AMR inequities; report incidental findings on equity dimensions of AMR. Articles that reported only on burden of disease and did not explore reasons for identified trends were excluded. As is common in CIS, the diverse nature of the types of studies included meant it was not appropriate to classify articles with standard quality assessment tools. "
	Methods	Was this review registered on Prospero or elsewhere?	No, we did consider this, but we did not register it on Prospero as it is not a sytematic review

	Introduction	In the Introduction, the authors state intersectional approaches are needed but are fairly new in the AMR field. They say this represents a missed opportunity to better tailor AMR interventions. It is important that the author share concrete examples of how an intersectional approach has informed interventions to improve programmatic effectiveness, equity, and sustainability for other disease outcomes. Otherwise, this review could be construed as more of a thought exercise rather than a paradigm shift.	We have added a section to the introduction that now reads: "Intersectionality enhances understanding of not only who is vulnerable but also the complex processes underlying these risks (23) and therefore informs policies, programmes and services to address these drivers of ill-health (20). For example, intersectional approaches have identified synergistic effects of age, ethnicity and gender (for example) that have led some to experience less positive impacts from health interventions than others or greater barriers to accessing health services during pandemics (28,29)" We have also clarified that a focus not just on intersectionality but on equity more broadly is relatively new to the AMR field: "However, a focus on equity and the use of intersectional approaches is fairly new in the AMR field "
		Lines 157-158: What resistant infections?	We have added the following: "Studies in Kenya, Brazil and Guinea-Bissau have identified that sex workers are at high risk of resistant infections such as Neisseria gonorrhoeae, Treponema pallidum and HIV-1 drug resistance, highlighting their occupational exposure to resistant infections (54–56)."
		Line 164: What mining chemicals? Susceptibility to TB or infection more broadly?	We have reworded this section for clarity to read: "Cuboia et al. (37) describe high rates of DR-TB in areas of Mozambique close to the South African border. They suggest that this is due to the presence of high numbers of people who undertake seasonal mining work in South Africa, which increases their exposure to mining chemicals (such as silica dust) and poorly ventilated working conditions, combining to increase the risk of TB (59)."

		Line 193-194: Sentence about vector borne disease, what does this have to do with AMR, specifically?	We have reworded this paragraph for clarity and it now reads: "Physical distance and lack of access to infection prevention methods can also impact mobile or migratory communities. Forest areas in Cambodia experience distinct malaria transmission characteristics due to specific forest vectors, climate and land use change, livelihoods, migration and poor health infrastructure (53). Those with forest-going occupations at the Thailand/Cambodia border (a current epicentre of the emergence of Plasmodium falciparum drug resistance) have limited access to mosquito nets and experience significant delays before they can access necessary antimicrobials (53). As a result, repeated delayed courses of antimicrobials, or self-prophylaxis for malaria, can increase the risk of resistance."
		Line 271: Shukla et al - is this in Tanzania, too? High level relative to urban dwellers?	We have added referenc to the study that we think is relevant from a Tanzanian context. This section now reads: "In rural Nepal, gender norms dictate that women are expected to be responsible as primary caregivers for knowing about how and when to give medicines to children (95), and similar norms were identified in Northern Tanzania (63)."
		line 230: What was found in China?	We have reworded this section to read: "Likewise, in northwestern China, antibiotic use for infection prevention in chicken farms was associated with lower income farms (81)."
		Line 304: Redwood et al paper - among whom?	We have edited this to read: "Redwood et al. (108) also report loss of reputation among people with DR-TB in Vietnam."

		A scoping review on the intersections between AMR pathogens and the SDOH was recently published in the Lancet Microbe. This would be relevant to include in introduction and/or discussion. PMID: 39653050.	Thank you for highlighting this. We have now included a reference to this in the introduction see introduction paragraph 1.
		Line 354: The authors say their review highlights how AMR intersects with caste, disability, refugee status... I see no mention of caste or disability status in their results. I see 1 reference to prescribers in refugee campus, not individuals' refugee status.	We intended to say that the review highlights the gaps in understanding on these intersections - as there was nothing on these intersections. We appreciate this wasn't clear so have edited this for clarity. It now reads : "Our review highlights significant knowledge gaps across LMIC contexts. We found no evidence that allowed us to draw inferences on how AMR intersects with ethnicity, caste, disability and very little on refugee status and urban informality."
		Line 152, 155: Why are diseases capitalized?	We have removed capitalisation
		Line 195-196: Shouldn't be a period before malaria	This has been removed

	Line 300: "This is echoes..." Sentence has poor grammar, also what findings?	This section has been edited and now reads: "Catastrophic costs were greater for those with DR-TB. Pham et al. (104) found that those with lower education and those unemployed at the beginning of MDR-TB treatment were more likely to face catastrophic costs, and for many income did not bounce back at the end of treatment. Likewise, in Zimbabwe, Timire et al. found that DR-TB narrowed job opportunities even after completion of treatment (105). In Brazil, this aspect of MDR-TB impacted those under 40 years most due to their breadwinning roles (102). Socioeconomic stressors also led to reduced psychological wellbeing (104,106). Taylor et al. (83) describe a "a vicious downward spiral in overall well-being, affecting most severely those who were already worse off to begin with" in South Africa and Uganda (p 8)."
	Line 306: as reported "among" HIV-positive...	This was reported by the migrants themselves as opposed to observed behaviours, so we feel that this language best reflects that
	Line 333: Doron et al citation is not in correct format	We have updated this citation
	Line 358: informal "or" pluralistic	This has been edited
	Lines 478-480 are two sentences but should be one sentence	This has been edited

4		The manuscript addresses relevant concerns related to the study of equity and antimicrobial resistance (AMR) through a critical interpretive synthesis (CIS) approach. While these are timely concerns, many of the issues discussed under the identified themes have already been discussed in previous CIS (see Pahlman et al., 2022, and Aguiar et al., 2024). A clearer positioning of the current manuscript in relation to existing CIS literature addressing AMR concerns would allow the authors to enrich the discussion and would enhance the added value of this manuscript.	Thank you for highlighting these other CIS, which are of interest and we agree are aligned with our arguments, though we note that our CIS is distinct - Pahlman et al focus on the framing and understanding of antimicrobials as a public good and of AMR as a security threat, and Aguiar et al have focused on the positioning of AMR-One health governance in relation to various framings and concepts. Both offer interesting insights into the positioning of AMR as an inequity or justice issue, and we have cited these articles and linked to them in the paper. In the introduction we situate our review in relation to these, stating: "Our review builds on recent CIS (35,36) that situate AMR policy and framings within wider frameworks of equity and justice, and which find that one health-AMR governance responses do not explicitly link integrate health equity concerns or considerations of the root causes driving AMR and antimicrobial use (AMU) spread and increase." We have also referenced key insights from these papers in the discussion.
---	--	--	--

		Even considering the readership and approach of Nature Communications, for a CIS engaging with critical theory, the article needs to be more rigorous in its approach to equity, (environmental) justice and intersectionality. In presenting a conceptual framework that is based on a CIS informed by critical theory, the manuscript would benefit from stronger critical grounding to inform a more engaged analysis of the structural inequities as they relate to AMR. A stronger engagement of intersectional theory across the relevant sections of the manuscript would also enable a deeper and theory-driven analysis of the structural inequities the authors themselves identify as key evidence gaps.	we have added more detail and reworded paragraphs in the introduction between pages 2 and 4, including more emphasis on value of intersectional lens to AMR, examples of value of use of intersectional lens in health more widely, and value of intersectionality to one health. We draw out the intersectional dimensions of our overarching findings in the 'summary of results' section. We have also brought out a more explicit reference to environmental justice in the introduction of the manuscript.
		The global dimensions of structural inequities are highlighted several times throughout the manuscript. This suggests that the authors recognize their importance for the conceptual framework and manuscript, however its importance is only briefly mentioned in the limitations section.	Thank you for highlighting this. We have added further mention of these throughout the paper including in the introduction (see paragraph 2), the conceptual framework description and the disussion (see paragraph 2, 7)

		The article offers a valuable synthesis of the evidence on structural inequities as they relate to AMR, but it would benefit from grounding the analysis of equity and (environmental) justice in conceptualizations aligned with critical or intersectional theory, a deeper and more rigorous engagement with intersectional frameworks, a relevant positioning within existing CIS literature, and a more substantive exploration of the global dimensions of AMR inequities, to reach its full potential.	Thank you - we have taken on board these thoughtful comments to strengthen the manuscript. Please see above for how we've addressed these.
--	--	--	---

REVIEWERS' COMMENTS

Reviewer #2 (Remarks to the Author):

Thank you for the opportunity to review changes to the manuscript. Concerns expressed in the review comments I provided have been addressed in the revisions. Thank you

Response to reviewer: Many thanks for your thorough and thoughtful review.

Reviewer #3 (Remarks to the Author):

The authors have adequately responded to most of my comments. Some points remain unclear; for example, I asked: "What data were actually extracted from each article?" The authors have shared that they used a data extraction framework that was "iteratively adapted" into several categories. It is still not clear what data were actually extracted from each article, regardless of the category that the article was binned into. Including a data extraction template as an appendix would be helpful.

Response to reviewer: Many thanks for your point of clarification. The iteratively developed framework included the following categories: susceptibility to infection, exposure to infection and antimicrobials, health-seeking pathways, self-medication and informal prescription, formal prescription practices, treatment completion/continuity, knowledge of AMR and antimicrobials, and experiences of care and impacts of infection (see p25). For each article, where relevant, quotations were extracted under each of these categories/themes. We subsequently used qualitative content analysis to synthesise qualitative findings within and across themes. As suggested, we have also added the above sentence to the Data extraction and analysis section on p25 to clarify. We have also included the data extraction template in the appendices.

I continue to have concerns with the subjective nature of this research approach. In response to my previously stated concerns, the authors wrote: "the rigour does not depend on its systematic or reproducible nature, but rather on its thorough interpretive approach and [that it is?] grounded in methods of qualitative inquiry." The authors have also clarified that articles could meet inclusion criteria even if they merely speculated on drivers of AMR inequities in their discussion, rather than rigorously investigated them. So, not only is the authors approach not meant to be reproducible, but the speculations of others were considered valid inputs to their synthesis of the literature..? Overall, I question how this approach might genuinely move the field forward or inform further debate as the authors have stated is their purpose.

Response to reviewer: Thank you for highlighting this - we appreciate that our wording wasn't clear here. All included articles reported findings related to equity, but some papers only contextualised these findings and explored the root causes or structural drivers of these in the discussion. We have simplified our language around the inclusion criteria in the manuscript to avoid confusion. This now reads.... "To

meet inclusion criteria, articles needed to describe findings related to how social or structural determinants of health impact on susceptibility or exposure to infection, transmission routes for AMR, access to treatment or the impact of the disease.” In some articles, when authors reflected on these equity findings in the discussion, they added further contextualisation and potential root causes of these in the discussion, which was valuable for our own analysis. A key factor in the choice of the CIS methodology is the fact that there is very little literature that does rigorously investigate the root causes of equity issues in AMR. Therefore, we feel that there is great value in synthesising the disparate knowledge in this area, and this includes drawing from authors’ discussion of their results in context. We hope that this edit in the manuscript simplifies and clarifies this point.

Reviewer #4 (Remarks to the Author):

The authors have effectively addressed all comments and concerns, and I believe the revised version will make a significant contribution to AMR research.

Response to reviewer: Many thanks for your thorough and thoughtful review.